# Functional Characterization and Anti-Tumor Effect of a Novel Group II Secreted Phospholipase A_2_ from Snake Venom of Saudi *Cerastes cerates gasperetti*

**DOI:** 10.3390/molecules28186517

**Published:** 2023-09-08

**Authors:** Mona Alonazi, Najeh Krayem, Mona G. Alharbi, Arwa Ishaq A. Khayyat, Humidah Alanazi, Habib Horchani, Abir Ben Bacha

**Affiliations:** 1Biochemistry Department, Science College, King Saud University, P.O. Box 22452, Riyadh 11495, Saudi Arabia; moalonazi@ksu.edu.sa (M.A.); mgalharbi@ksu.edu.sa (M.G.A.); aalkhyyat@ksu.edu.sa (A.I.A.K.); hialanazi@ksu.edu.sa (H.A.); 2Laboratoire de Biochimie et de Génie Enzymatique des Lipases, ENIS, Université de Sfax, Route de Soukra 3038, Sfax BP 1173, Tunisia; krayemnajeh@yahoo.fr; 3Science Department, College of Rivière-Du-Loup, Rivière-Du-Loup, QC G5R 1R1, Canada; habib.horchani@cegeprdl.ca

**Keywords:** snake venom, secreted phospholipase A_2_, stability, anti-tumor effect, cytotoxicity, human cancer cell lines

## Abstract

Secreted phospholipases A_2_ are snake-venom proteins with many biological activities, notably anti-tumor activity. Phospholipases from the same snake type but different geographical locations have shown similar biochemical and biological activities with minor differences in protein sequences. Thus, the discovery of a new phospholipase A_2_ with unique characteristics identified in a previously studied venom could suggest the origins of these differences. Here, a new Group II secreted phospholipase A_2_ (Cc-PLA_2_-II) from the snake venom of Saudi *Cerastes cerastes gasperetti* was isolated and characterized. The purified enzyme had a molecular weight of 13.945 kDa and showed high specific activity on emulsified phosphatidylcholine of 1560 U/mg at pH 9.5 and 50 °C with strict calcium dependence. Interestingly, stability in extreme pH and high temperatures was observed after enzyme incubation at several pH levels and temperatures. Moreover, a significant dose-dependent cytotoxic anti-tumor effect against six human cancer cell lines was observed with concentrations of Cc-PLA_2_ ranging from 2.5 to 8 µM. No cytotoxic effect on normal human umbilical-vein endothelial cells was noted. These results suggest that Cc-PLA_2_-II potentially has angiogenic activity of besides cytotoxicity as part of its anti-tumor mechanism. This study justifies the inclusion of this enzyme in many applications for anticancer drug development.

## 1. Introduction

The horned viper *Cerastes cerastes*, which belongs to the Viperidae family, is found in numerous deserts, particularly in Jordan, Egypt, Tunisia, Algeria, and Saudi Arabia [1]. *Cerastes cerastes gasperetti* is the most common snake in the Saudi Arabian desert, alongside the black cobra (*Walterinnesia aegyptia*), the Arabian cobra (*Naja haje arabica*), and *Walterinnesia morgana* [2].

Viperidea venom contains various pharmacologically active molecules, especially small peptides and proteins, which can be classified into two major families: enzymes including serine proteinases, zinc-dependent metalloproteases, and secreted phospholipases A_2_, and proteins with no enzymatic activity, such as C-type lectin-like proteins and desintegrins [3].

Phospholipase A_2_ (PLA_2_) are among the protein compounds of viperidea venom and represents more than 10% of the dry weight [4]. These enzymes hydrolyze ester bonds at the sn-2 position in membrane glycerophospholipids, which produce unsaturated free fatty acids and lysophospholipids [5]. Based on primary sequences, molecular weight, number and position of disulfide bonding, and the requirement of calcium for hydrolysis, PLA_2_ is assigned to five main categories: Ca^2+^-independent (iPLA_2_), cytosolic Ca^2+^-dependent (cPLA_2_), secreted (sPLA_2_), platelet-activating factor (PAF) acetylhydrolase and lysosomal PLA_2_ [6].

Snake-venom PLA_2_ belongs to sPLA_2_ with a low molecular weight (14–18 kDa), which are cross-linked by five to eight disulfide (S-S) bonds and a histidine/aspartate (His/Asp) catalytic dyad. Based on their primary structural homology and disulfide bonding pattern, snake venom sPLA_2_ is classified into two groups: I and II [6]. Group II is found in viperidea venom, while group I characterizes Elapidae and Hydrophidae snake venom [7]. These groups can be further divided into subgroups—active Asp49-PLA_2_ and Lysine 49-PLA_2_—which are catalytically inactive due to their inability to bind calcium [8]. A structural comparisons of snake venom sPLA_2_ shows that they share 40 to 99% identity and have significant similarity in their three-dimensional structure [9]. Despite this similarity, these sPLA_2_ exhibit, independent of their catalytic phospholipid hydrolysis, pharmacological activities, including hemolytic, anticoagulant, and antiplatelet activities, as well as neurotoxic, myotoxic, antiviral, and bactericidal properties [10]. According to Kini et al., these activities are mediated by the interactions between sPLA_2_s and receptors localized in target cell plasma membranes through pharmacological sites that are separate from the catalytic sites [10].

The anti-tumor activity of snake venom is among the best-described and studied biological effects of sPLA_2_. These enzymes, which associated or not with toxins, are described to be responsible for tumor regression through cytotoxicity against human cancer cells [8]. Indeed, sPLA_2_ isolated from *Bothrops asper*, *Bothrops moojeni*, *Naja naja*, and *Walterinnesia aegyptia* snake venom showed cytotoxicity against mouse adrenal tumor cells, Ehrlich ascites tumor, human breast adenocarcinoma, leukemia, and colorectal carcinoma [7,8]. sPLA_2_ from Russell’s viper venom altered melanoma tumor cell adhesion in vitro, in turn affecting tumor mass growth in vivo [11]. The anti-tumor effect of sPLA_2_ manifested as inhibited proliferation of endothelial cells through interaction with vascular endothelium growth factor (VEGF) receptor-2, which is involved in angiogenesis [12].

Various isolated sPLA_2_ from the venom of the same types of snakes from different regions showed differences in biochemical and/or biological properties. For example, myotoxins with PLA_2_ activity purified from *Bothrops asper*, which cause the majority of human envenoming in Central American, exhibited various effects on human cells [13]. Another differences is observed with phospholipases A_2_ purified from the venom of Indian cobra species such as *Naja naja naja*, which is distributed across the Indian subcontinent and *Naja naja kaouthia*, which is localized in northeastern India, Bangladesh, Myanmar, and most of Southeast Asian countries [14]. These PLA_2_ were found to decrease the viability of mouse melanoma cell line B16-F10 [15], showed anti-invasive and anti-metastatic activity against human breast-cancer cells [16], and were cytotoxic against human breast-cancer MCF7 cells and lung cancer A549 cells [17]. Along the same lines, two secreted PLA_2_ purified from the venom of Tunisian *Cerastes cerastes* presented an anti-tumor effects without cytotoxicity by blocking the adhesion, migration, and invasion of human endothelial cells by inhibiting αv and α5β1 integrins. Moreover, from the same snake in the Algerian desert, two sPLA_2_ were purified and characterized, and showed different biological and biochemical properties compared to the sPLA_2_ from the Tunisian *C. cerastes* [4,18,19].

The existence of protein isoforms belonging to the same class in the venom of the same snakes from various geographic locations could be related to mutations in the corresponding encoding genes during evolution [20,21]. Although they share a similar primary structure, these isoforms have different biological targeting and physiological effects. This particularity encourages their use for structure–function correlations, allowing the design of novel anti-toxic drugs for the treatment of severe envenomation and diseases. Also, the purification and characterization of new components from the same snake might help us to better understand the pathophysiology of envenomation, and thus discover new molecules that could be useful for investigating the molecular processes of disease [3].

This study reports the functional characterization of a new venom-secreted PLA_2_ purified from horned viper *Cerastes cerastes gasperetti* (Cc-PLA_2_-II) from the Saudi Arabian desert. Interesting anti-tumor effects against six human tumor cell lines were observedand compared to those of sPLA_2_ from venom of *Cerastes cerastes* from the Tunisian desert. The biological activity of snake venom sPLA_2_s suggests that they may have a crucial role in altering tumor progression, and therefore could be potential targets for new anticancer drugs.

## 2. Results and Discussion

### 2.1. Cc-PLA_2_-II Purification

Cc-PLA_2_-II was purified according to the protocol described in Section 3. After heat treatment of crude Cerastes cerastes gasperettii venom at 75 °C for 10 min, ethanol fractionation, followed by a reverse-phase high-pressure liquid chromatography (RP-HPLC), were performed. The peak retention time of 17 min, corresponding to 50% acetonitrile elution, showed the pure Cc-PLA_2_-II with phospholipase activity tested under standard conditions (Figure 1A). Cc-PLA_2_-II was purified with a recovery of 24.3% and a purification factor of 34.59 (Table 1). A single band in the SDS-PAGE profile indicated an apparent molecular mass of 14 kDa (Figure 1B). The molecular mass was precisely determined by MALDI-TOF analysis, showing a single peak with a molecular mass of 13.945 kDa (Figure 1C).

Using emulsified phosphatidylcholine (PC) at 13 mM as the substrate, purified Cc-PLA_2_-II presented a specific activity of 1560 U/mg; this is different from other snake-venom sPLA_2_ purified from the venom of Tunisian and Algerian *Cerates cerastes* venoms, which had values of 51.5 U/mg [18] and 13,333 U/mg [9], respectively. This difference may be related to differences in the PLA_2_ interfacial recognition domain composed of hydrophobic residues responsible for binding enzyme and therefore hydrolyzing substrate. Indeed, it is well known that reactions at the interface by lipolytic enzymes such as PLA_2_ can be divided into two steps: interfacial recognition, followed by catalytic hydrolysis [22]. The interfacial binding surface of sPLA_2_s involves 10 to 20 amino acid residues, which are different between sPLA_2_ [23]. A comparable level of specific activity was noted between Cc-PLA_2_-II and WaPLA_2_-II from *Walterinnesia aegyptia* venom [7] and LM-PLA_2_-II from *Lachesis muta* venom (1250 and 1428 U/mg, respectively) [24].

**Table 1 molecules-28-06517-t001:** Characterization of CcPLA_2_-II at each purification step.

Purification Step	Total Activity ^1^ (U)	Protein ^2^ (mg)	Specific Activity (U/mg)	Activity Recovery (%)	Purification Factor
Extraction	9250	205	45.1	100	1
Heat treatment (75 °C, 10 min)	7030	58	121.2	72	2.69
Ethanol fractionation (50–90%)	4950	11.2	441.9	53.5	9.8
C18 RP-HPLC	2250	1.44	1560	24.3	34.59

^1^ One unit of phospholipase activity was defined as 1 μmol of fatty acid liberated under assay conditions. ^2^ Protein concentration was measured using the Bradford method [25].

Through Edman degradation, the first 44 amino acid residues from the Cc-PLA_2_-II *N*-terminal sequence were determined. The obtained sequence was: _NH2_ QFGKMIKLTTGKSAALSYSAYGCYCGWGGQGKPQDATDHCCFLH _COOH_. 

As shown in Table 2, Cc-PLA_2_-II demonstrates remarkable homology (≥85%) with known amino acid sequences of group II sPLA_2_ from the venom of *Cerastes cerastes* [26], *Vipera renardi* [27], and *Vipera ursinii* [28]. According to the alignment of primary sequences, Cc-PLA_2_-II belongs to Group II sPLA_2_, like all phospholipases isolated from viperidea venom [29]. However, this result should be confirmed by further determining the total protein sequence. Two isoforms of sPLA_2_ from Tunisian *Cerastes cerastes* venom were purified and sequenced designed by CC-PLA_2_-1 and CC-PLA_2_-2 [26]. Another isoform werewas isolated from Algerian of the same snake species from Algeria [18]. Based on a study of venom from *Cerastes cerastes* from Saudi Arabia, the particular composition of this venom could be attributed to geographical repartitioning or gene mutations, which would explain the observed polymorphism in PLA_2_ sequences. 

### 2.2. Cc-PLA_2_-II Biochemical Characterization

#### 2.2.1. Enzymatic Activity

The enzymatic activity of Cc-PLA_2_-II was determined using the chromogenic substrate, 4-nitro-3-octanoyloxybenzoic acid (4N3OBA), as described in the Section 3. With the same concentrationof this substrate, Cc-PLA_2_-II, exhibited the highest activity with Michaelian behavior (677.2 nmol/mg/min) compared to the positive control (commercial PLA_2_ from *Naja mossambica mossambica* Nm-PLA_2_, 333.6 nmol/mg/min) and Wa-PLA_2_-II (481.6 nmol/mg/min) (Table 3). The ability of Cc-PLA_2_-II to hydrolyze the chromogenic substrate suggests that this enzyme belongs to active snake-venom PLA_2_ with an aspartate residue at position 49, as seen previously in sPLA_2_ from *C. durissus collilineatus* and PrTXIIIB from *B. pirajai* [30]. Indeed, snake-venom PLA_2_, in which Asp 49 is replaced by a lysine residue 49, such as BnPS-7 from *Bothrops pauloensis* [31], Basp-III from *B. asper* [32], and BthTXI from *B. jararacussu* [33], are unable to hydrolyze phospholipid substrate or 4N3OBA.

According to the kinetic behavior, Cc-PLA_2_-II was classified as a typical sPLA_2_, since it hydrolyzes a synthetic substrate at position *sn-2* and is normally able to attack substrates in a micellar state, which was confirmed by its specific activity of 1560 U/mg on emulsified PC substrate at 13 mM (Table 1). Many snake-venom sPLA_2_s, such as BrTX-I isolated from *Bothrops roedingeri* [34], trimorphin from *Trimorphodonbiscutatus lambda* [35], and AplTx-I from *Agkistrodon piscivorus leucostoma* [36], present the same behavior. Indeed, Cc-PLA_2_-II has a specific activity of 1560 U/mg on emulsified PC substrate (Table 1). The high activity compared to other venom phospholipases can be explained by the higher efficiency of interfacial catalysis supported by its high adsorption to the lipid–water interface enhanced by the presence of anionic amphipathic amino acids residues [34].

#### 2.2.2. Effect of pH and Temperature on Activity and Stability

The activity and stability of Cc-PLA_2_-II in response to thephysicochemical parameters pH and temperature were investigated (Figure 2). Although almost of snake-venom PLA_2_ substrates undergoes hydrolysis have acidic pH, Cc-PLA_2_-II exhibited the highest enzymatic activity at alkaline pH of 8 and 9 (Figure 2A). This property was also observed with other snake-venom sPLA_2_, such as trimorphin from the venom of *Trimorphodon biscutatus lambda*, which is fully active at pH between 7 and 9 [35], and Wa-PLA_2_-II from *W. aegyptia* snake venom, with optimum activity at pH 9.5 [7].

Moreover, the pH stability study showed that Cc-PLA_2_-II was quite stable over a large pH range, conserving its total activity when incubated at a pH ranging from 5 to 9 and more than 90% of its original activity at pH 4 or 11 (Figure 2B). For Cc-PLA_2_-II, 75% of its initial activity was maintained at pH 12. The same stability profile was also observed with Wa-PLA_2_-II, for which total activity was maintained between pH 6 and 10, and less than 40% of the original activity was lost at pH 12 or 4 [7]. Cc-PLA_2_ purified from the venom of Tunisian *Cerastes cerastes* were also stable at pH from 3 to 9 [37]. Yuan et al. explained the stability at extreme pH through an RMN study of porcine PLA_2_. The tertiary structure of these PLA_2_s became less ordered at acidic and alkaline pH, but rigid interactions still existed between helices [38].

The optimum enzymatic activity of most snake-venom sPLA_2_s was determined at one temperature (37 °C). A few studies describe the effect of temperature on enzymatic activity or stability. In this regard, interestingly, Cc-PLA_2_-II displayed a high catalytic activity at 50 °C (Figure 2B). This activity at high temperatures has also been observed with the same snake-venom sPLA_2_, such as CC-PLA_2_ from Tunisian *Cerastes cerastes*, which is fully active at 60 °C [9], and Wa-PLA_2_-II [7] and PLA_2_ from *Austrelaps superba* [39], with maximum activity at 55 °C and 50 °C, respectively. According to Iyer et al., thermophilic and mesophilic proteins have similar three-dimensional structures but are different in their active site residues. Most of thermophilic enzymes have arginine and proline residues and less asparagine/glutamine content compared to mesophilic ones [40]. This was confirmed in an analysis of the primary structure of thermophilic CC-PLA_2_ (active at 60 °C) and mesophilic MVL-PLA_2_ from *Macrovipera lebetina* (active at 37 °C) [9]. Indeed, thermo-enzymes have a conformational three-dimensional structure that is more rigid with higher packing efficiency, reduced unfolding entropy, and a-helix stability of maintained by many interactions, such as hydrogen bonds, electrostatic and hydrophobic interactions, and disulfide bonds [40].

Interestingly, Cc-PLA_2_-II was still fully active after incubation at temperatures ranging from 20 to 55 °C. Even after treatment at 80 °C and 90 °C for 60 min, only 54.5 and 24.5% of maximum activity, respectively, was maintained (Figure 2B). This loss of enzymatic activity may be due to protein denaturation, during which the enzyme tertiary structure is unfolded at high temperature into a disordered polypeptide in which key residues that participate in functional or structural stabilizing interactions are no longer closely aligned [40].

Thermostability was also observed with MVL-PLA_2_ from *Macrovipera lebetina*, CC-PLA_2_ from Tunisian *Cerastes cerastes,* and Wa-PLA_2_-II. About 71 and 45% of maximum activity was preserved in MVL-PLA_2_ after incubation at 60 °C and 90 °C, respectively. CC-PLA_2_ is less stable, and 93 and 54% of its maximum activity was maintained after incubation at 60 and 90 °C, respectively [9]. Wa-PLA_2_-II still showed 74 and 45% of residual enzymatic activity at 65 and 70 °C, respectively. After treatment for 60 min at 80 and 90 °C, 50% of maximum activity was retained in Wa-PLA_2_-II [7]. Enzyme stability at high temperatures can explained by long-term or kinetic stability and thermodynamic or conformational stability. The latter concerns the resistance of the folded enzyme conformation to denaturation maintained by its rigidity and composition of hydrophobic residues. Long-term stability, expressed as its half-life, measures the resistance to irreversible inactivation. Molecular flexibility allows efficient catalysis and stability requires extra interactions, and rigidity-characterizes enzyme that are stable at high temperatures [40].

#### 2.2.3. Bile Salt, Calcium, and Metal Ion Dependence

Tensioactive agents such as bile salts play a key role in the dispersion of hydrolysis products of lipolytic enzymes such as lipases and phospholipases. Therefore, the effect of bile salts on Cc-PLA_2_-II activity was investigated. Under optimal conditions, Cc-PLA_2_-II enzymatic activity can be determined using different concentrations of two bile salt sodium taurodeoxycholate (NaTDC) and sodium deoxycholate (NaDC) (0 to 10 mM). Maximal activity was achieved with 6 mM bile salt (Figure 3A).

MVLPLA_2_ and Cc-PLA_2_ exhibited maximum phospholipase activity in the presence of 1 and 3 mM NaTDC, respectively. Above these concentrations, phospholipase activity decreased progressively [9]. Wa-PLA_2_-II showed the highest activity with 4 mM NaTDC and 6 mM NaDC [7]. Dependence of PLA_2_ on bile salts was explained by Pan et al., who suggested that PLA_2_ activity on PC emulsion increases in the presence of added bile salts since the anionic charge of the amphiphilic bile salt converts an otherwise zwitterionic interface to an anionic interface, which is more suitable for the catalytic action of pancreatic secreted PLA_2_ [41].

It is well known that calcium ion (Ca^2+^) is crucial for binding to the substrate and for catalysis of sPLA_2_ [42]. In this regard, using emulsified PC as a substrate, Cc-PLA_2_-II enzymatic activity was measured in the presence of various concentrations of Ca^2+^. Like CC-PLA_2_, MVLPLA_2_, and WaPLA_2_-II, which exhibited millimolar calcium-dependent enzymatic activity of 1, 3, and 8 mM, respectively [7,9], Cc-PLA_2_-II showed stricter maximum phospholipase activity at 8 mM (Figure 3A). Indeed, Cc-PLA_2_-II was completely inactive without calcium or when incubated in the presence of a cation chelator such as ethylenediaminetetraacetic acid (EDTA) or ethylene glycol-bisaminoethyl ether tetraacetic acid (EGTA) (Figure 3A).

Crystal structure studies of snake-venom sPLA_2_ have shown that the calcium-binding loop is the most conserved region, and it extends from tyrosine 25 to glycine 33 localized in a loop in the first *N*-terminal helix H1. Therefore, calcium ion is required to provide the final shape to this loop, which lends support to the substrate-binding site [43]. The calcium ion is coordinated by two carboxylate oxygen atoms of aspartate residue (at position 49 in PLA_2_ groups I and II and 34 in group III) and a disulfide bond between two cysteine residues on the H1 helix, helping to maintain calcium position and provide conformational stability to the otherwise flexible loop. This loop forms the wall of the hydrophobic channel of the substrate, confirming its participation in stabilizing the intermediate tetrahedral transition state achieved during catalysis [44].

The effect of 10 mM metal ions (Cd^2+^, Hg^2+^, Mg^2+^, and Mn^2+^) in the absence or presence of 1 and 10 mM Ca^2+^ on Cc-PLA_2_-II activity was investigated. PLA_2_ activity was measured under optimal conditions (pH 9 and 55 °C) (Figure 3B). As seen with other sPLA_2_, such as AplTx-I from *Agkistrodon piscivorus leucostoma,* in the absence of Ca^2+^, positive metal ions were unable to activate Cc-PLA_2_-II, indicating that the substitution of Ca^2+^ by other ions significantly decreases PLA_2_ activity [36]. However, the addition of 1 mM Ca^2+^ with 10 mM of metal ions increased enzymatic activity by up to 50%. This kinetic behavior toward divalent cations confirmed the strict dependence of Cc-PLA_2_-II enzymatic activity on Ca^2+^. It plays an important role in both binding and catalytic activity and in the stabilization of the calcium-binding loop, to optimize the enzymatic conformation during interactions with the substrate [42]. Its replacement by other cations with the same valence as calcium abolished the phospholipase activity, suggesting that it is unable to bind to the substrate or that cause catalytic inhibition probably due to the size and number of electron shells [45].

### 2.3. Effects on Biological Membranes

#### 2.3.1. Hemolytic Effect

It is well documented that snake-venom sPLA_2_ causes cytotoxic effects and damage biological membranes by hydrolyzing the phospholipids of the bio-membranes and/or generating lysophospholipid and free fatty acids as well as phospholipid hydrolysis products, that are lytic and cause considerable membrane damage [46]. In this regard, the hemolytic effect of Cc-PLA_2_-II was investigated, since erythrocyte membranes are among the best-studied biological materials. PC and sphingomyelin are the major phospholipids present in these membranes [47]. Cc-PLA_2_-II exhibited only indirect hemolytic activity of up to 90% at a concentration of 0.75 nM after 30 min incubation with human erythrocytes. At the same concentration, Nm-PLA_2_ and Wa-PLA_2_ showed hemolytic activity of 83.5 and 52%, respectively (Figure 4). This behavior was observed with various snake-venom sPLA_2_ that present both direct and indirect hemolytic activity. Indeed, 10 µg of NK-PLA_2_-I showed 25.2% of indirect hemolytic activity. Indirect hemolysis can be explained by the formation of phospholipid hydrolysis products (lysophospholipids and free fatty acids) inducing hemolysis [46].

#### 2.3.2. Cytotoxic Effect

The cytotoxic effect of purified Cc-PLA_2_-II in concentrations from 0.25 to 8 µM toward human cancer cell lines (MDA-MB-231, MCF-7, HCT-116, HT-29, NCI-292, and DR) was determined via MTT assay have been previously described in the Section 3. After 24 h of treatment with 1 µM of Cc-PLA_2_-II, the percentage of viable cells was about 50% for MDA-MB-231, MCF-7, HCT-116, and DR, and nearly 25% for HT-29 and NCI-292 (Figure 5). An IC_50_ value of 3 µM was recorded for MDA-MB-231, MCF-7, HCT-116, and RD (Figure 5A–D). However, NCI-292 and HT-29 showed the highest sensitivity, with IC_50_ values of 1.8 and 2.2 µM, respectively (Figure 5E,F). The fact that cell damage is generally mediated by a specific interaction of PLA_2_ with cell surface receptors that differ among various cancer cells could explain the difference in sensibility between the cancer cell lines [48].

Several snake-venom sPLA_2_s showed anti-tumor activity through cytotoxic effects against various human cell lines. BthTX-I, isolated from *B. jararacussu*, presented IC_50_ values of 6.8 and 8 µM against human breast-cancer SK-BR-3 and MCF-7 cells, respectively [49]. An acidic PLA_2_ from *Bothrops moojeni* venom showed cytotoxic potential against human breast-cancer cells MDA-MB-231 with an IC_50_ of about 409 µg/mL [49]. Crotoxin from *Crotalus durissus terrificus* venom presented a heterogeneous response profile against human pancreatic, esophageal, cervical, and glioma cancer cell lines, with IC_50_ values ranging from 0.5 to 4 µg/mL. Like Cc-PLA_2_-II, crotoxin does not reduce the viability of normal cells. Significant cytotoxic activity against colon adenocarcinoma (HT29), melanoma (B16F10), and breast adenocarcinoma (MCF-7) cells has been reported, with IC_50_ values of 40, 108.3, and 308.6 µg/mL, respectively [48]. Four isoforms of a PLA_2_ from venom of *E. ocellatus*, *E. coloratus*, *E. carinatus sochureki*, and *E. pyramidum* exhibited a cytotoxic activity against lung adenocarcinoma A549 with IC_50_ values of 5.2, 3.5, 8.5 and 2.9 µM, respectively [50].

Cc-PLA_2_-II did not show any cytotoxic effect on the viability of human umbilical-vein endothelial cells (HUVECs) after 24 h, at concentrations ranging from 0.25 to 8 µM (Figure 5G). This result agrees with previous work suggesting that snake-venom sPLA2 specifically target cancer cells without affecting normal cells [51] and Cc-PLA2-II undergoes selectivity toward cancer cells compared to normal cells, suggesting its potential anti-angiogenic activity. This selectivity also suggests a difference in cell membrane composition, especially in phospholipid domains between cancer and normal cells [51,52].

Compared to almost all types of snake-venom Asp49-sPLA_2_, which are cytotoxic to normal and cancer cells, Cc-PLA_2_-II selectivity is a new property that allows us to use it in cancer investigation by targeting several signaling pathway. Indeed, studies have demonstrated that nontoxic snake-venom PLA_2_ such as CC-PLA_2_ and MVL-PLA_2_ exhibited an anti-angiogenic effect on cancer cells and endothelial cell lines. At concentrations up to 250 nM, MVL-PLA_2_ did not significantly affect the viability of the HMEC-1 endothelial cell line without inducing membrane damage determined by the release of LDH. The same results were observed with CC-PLA_2_-1 and CC-PLA_2_-2 at 1 µM using HBMECs. These sPLA_2_s inhibited endothelial cell adhesion, migration, and invasion by impairing cell-extracellular matrix interactions through blocking integrin receptors α5β1 and αvβ3 [26,51]. Along the same lines, CC-PLA_2_-II could be very useful for inhibiting tumor-associated vasculature and could be a relevant target for inhibiting tumor progression by blocking angiogenesis steps and stopping tumor progression through its cytotoxicity against only cancer cells.

Generally, snake-venom PLA_2_ exhibit a time- and dose-dependent cytotoxicity against various cancer cells without any effects on normal cells, suggesting that they have selective toxicity against tumors. Studies have described that this anti-tumor activity is related to interactions between the C-terminal region of venom PLA_2_ and cell membranes. Synthetic peptides derived from this region can disrupt hydrophilic membrane cells, creating pores, allowing the penetration of the peptides or PLA_2_ into the intracellular medium [53].

Regarding cytotoxicity, several mechanisms have been demonstrated using with the use of flow cytometry to analyze the cell-cycle research of cell-death pathways [54]. the cytotoxicity of PLA_2_ may be related to the disruption of mitotic cell progression, which would imply an important anti-tumor mechanism, as evaluated by flow cytometry experiments. As illustrated by Silva et al, apoptosis and necrosis are the two mechanisms of cytotoxicity induced by snake-venom sPLA_2_ against several types of tumor cells. The apoptosis-inducing ability can be assessed by flow cytometry using annexin V conjugated to fluorescein, which acts as a marker for the phosphatidylserine exposed on cells, and a necrotic effect can be evaluated by staining treated cells with propidium iodide (PI), which marks the nucleic acid of dead cells [53].

The apoptosis mechanism induced by sPLA_2_ can be triggered by changes in mitochondrial membrane potential, cytochrome C release, and caspase-3 pathway activation. Some snake-venom PLA_2_s, such as CTX from *Crotalus durissus terrificus* venom, induce cell death by activating the autophagy in breast tumor cells [48].

Another mechanism was established by Marcussi et al. [55], who suggested that snake-venom sPLA_2_s might exert its a genotoxic effect by permanent DNA damage and micronuclei formation as shown in human lymphocytes evaluated by comet assay. Therefore, sPLA_2_s cause micronuclei formation indicating DNA breakage and/or aneuploidy induction impairing cell-cycle division [55]. In addition, cytotoxicity caused by these sPLA_2_s might involve liberated reactive oxygen species during its membrane phospholipid degradation, allowing increased intracellular oxidative stress by the activation of cell-death pathways and the downregulation of anti-apoptotic proteins [49]. It can be concluded that the cell cytotoxicity mechanism induced by snake-venom PLA_2_s depends on tumor type, which may indicate the distinct mechanism of action.

## 3. Material and Methods

### 3.1. Reagents

Chemicals were obtained from commercial sources. Chromatography material (reverse-Phase high-Performance Liquid Chromatography, (RP-HPLC)), phosphate buffered saline (PBS, pH 7.4), ethanol, acetonitrile, trifluoroacetic acid (TFA), sodium dodecyl sulfate (SDS), acrylamide, ammonium persulfate, *N*,*N*,*N*′,*N*′-tetramethyl ethylenediamine (TEMED), β-mercaptoethanol and coomassie brilliant blue R-250 were obtained from Bio-Rad (Hercules, CA, USA).

PC, 4-nitro-3-octanoyloxybenzoic acid (4N3OBA), NaCl, CaCl_2_, Tris-HCl, bovine serum albumin (BSA), sodium taurodeoxycholate (NaTDC), sodium deoxycholate (NaDC), ethylenediaminetetraacetic acid (EDTA), Triton X_100_, protein markers for molecular mass, and commercial PLA_2_ from *Naja mossambica mossambica* (*Nm*-PLA_2_, Sigma P7778) were purchased from Sigma Aldrich (St. Quentin-Fallavier, France).

Dulbecco’s Modified Eagles Medium, fetal calf serum, glutamine, streptomycin, penicillin, and 3-[4,5-dimethylthiazol-2-yl]-2,5 diphenyl tetrazolium bromide (MTT) were purchased from Life Technologies, Paisley, UK.

### 3.2. Purification of Cc-PLA_2_-II

About 30 speciments of *C. c. gasperettii* snakes were collected from Saudi Arabia’s central area and milked to extract about 300 mg of crude venom. After being dissolved in 0.2 M PBS (pH 7.4), the crude venom was heated at 75 °C for 10 min, then fractionated by precipitation with 50–90% (*v*/*v*) ethanol and finally applied to a reverse-phase high-performance liquid chromatography (RP-HPLC) C18 column (5 µm, 250 mm × 4.6 mm) pre-equilibrated with 0.1% TFA in water. A linear acetonitrile gradient (0–100%) allowed the elution of adsorbed proteins at a flow rate of 0.8 mL/min over 35 min.

### 3.3. Enzymatic Characteristics of Cc-PLA_2_

Using PC as substrate, PLA_2_ activity in the presence of calcium and sodium deoxycholate (NaDC) and at the indicated temperatures and pH values was determined by titrimetric pH stat assay according to Abousalham and Verger [56]. One μmole of fatty acid liberated under optimum pH and temperature conditions corresponds to one unit of phospholipase activity. Pure Cc-PLA_2_ was also tested on 4-nitro-3-octanoyloxybenzoic acid (4N3OBA) as described by Serino-Silva et al. [57]. Briefly, using a 96-well plate, 200 µL of buffer A (10 mM Tris-HCl buffer, pH 7.8, including 100 mM NaCl and 10 mM CaCl_2_), was added to each well. Thereafter, 20 µL of the enzyme solution, 20 µL of the substrate (1 mg/mL of 4N3OBA dissolved in acetonitrile), and 20 µL of water were added. Wa-PLA_2_-II from *Walterinnesia aegyptia* [7] and commercial PLA_2_ from *Naja mossambica mossambica* (*Nm*-PLA_2_, Sigma P7778) were used as positive controls, while non-enzymatic BSA was used as negative control. The reactions were performed in triplicate and was carried out at 37 °C for 150 min, and readings were recorded at 425 nm at 10 min intervals.

### 3.4. Protein Analysis

Protein quantitation was estimated using the Bradford method [25], while the purity and the molecular weight of the purified PLA_2_ were evaluated by the SDS-PAGE using 15% polyacrylamide gel in the presence of β-mercaptoethanol as a reducing agent [58]. The molecular mass of the native protein was then accurately determined on a Voyager DE-RP MALDI-TOF mass spectrometer (Perseptive Biosystem, Framingham, MA, USA). Mass spectra recorded in the linear mode were externally calibrated with suitable standards and analyzed by GRAMS/386 software. The *N*-terminal sequencing of the native CC-PLA_2_ was determined by automated Edman degradation according to Hewick et al. [59] using the Procise system (Applied Biosystems, Foster city, CA, USA). *Para*-Bromophenacyl bromide (*p*-BPB) was used for the chemical modification of Cc-PLA_2_ as described by Condrea et al. [60].

### 3.5. In Vitro Hemolytic Activity

The protocol described by Boman and Kaletta was followed to evaluate the direct and indirect hemolytic potency of the purified PLA_2_ [61]. To test the direct hemolytic activity, mixtures of varying enzyme amounts (0 to 3 μg) and substrate suspensions (1 mL of washed erythrocytes in PBS) were incubated for 30 min at 37 °C. Prior to that, a mixture of the substrate emulsion (8 mL) and fresh PC (1 mL) was prepared to measure the indirect hemolytic activity. The reaction was stopped by adding ice-cold PBS (9 mL) followed by centrifugation at 1500× *g* for 20 min at 4 °C). The proportion of hemoglobin (Hb) liberated at 530 nm against 100% hemolysis produced by adding Triton X-100 (1%) was used to quantify the hemolysis degree caused by CC-PLA_2_.

### 3.6. Cytotoxicity by MTT Assay

Human colorectal carcinoma (HT-29 and HCT-116), breast adenocarcinoma (MDA-MB-231 and MCF-7), lung mucoepidermoid carcinoma (NCI-H292), rhabdomyosarcoma (RD), and umbilical-vein endothelial cell (HUVEC) lines were obtained from the American Type Culture Collection (ATCC, Manassas, VA, USA). All cells were grown in Dulbecco’s Modified Eagle Medium supplemented with 10% fetal calf serum, 1% glutamine, 1% streptomycin, and 1% penicillin in 5% CO_2_ at 37 °C.

Cells were used to study the anticancer potency of the purified Cc-PLA_2_ by evaluating their viability using the MTT assay according to a protocol adapted from Mosmann (1983) [62]. The cells were plated in 96-well microplates (4 × 10^4^ in each well) and incubated for 24 h with varying concentrations of Cc-PLA_2_ (0.25 to 8 µM). Culture medium was used as a negative control. After removing the supernatant, 0.1 mL of MTT solution (1 mg/mL dissolved in culture medium) was added to the cells and incubated for 4 h. The resulting formazan crystals were resuspended with PBS (0.1 mL), and the optical density (OD) was measured at 550 nm. Cell viability was calculated as follows: Cell viability (%) = (OD_Sample_ ÷ OD_negative control_) × 100. Experiments were carried out at least in triplicate.

### 3.7. Statistical Analysis

Data were expressed as mean ± standard deviation (SD) using Excel software. A student’s *t*-test was performed to determine the statistical significance of differential findings between control and experimental groups.

## 4. Conclusions

A new Group II snake venom-secreted PLA_2_ from the venom of the *Cerastes cerastes gasperettii* desert cobra from Saudi Arabia, designated as Cc-PLA2-II, was purified and characterized. Compared to other snake-venom Asp49-sPLA_2_, which are cytotoxic to normal and cancer cells, Cc-PLA_2_-II presented a novel, interesting heterogenic and selective anti-tumor profile against several human cancer cell lines in a dose-dependent manner. Further studies will be carried out to determine the cytotoxic mechanism of Cc-PLA_2_-II. These results add to the functional knowledge database on snake-venom PLA_2_ and provide new insight into the development of antitumor drugs.

## Figures and Tables

**Figure 1 molecules-28-06517-f001:**
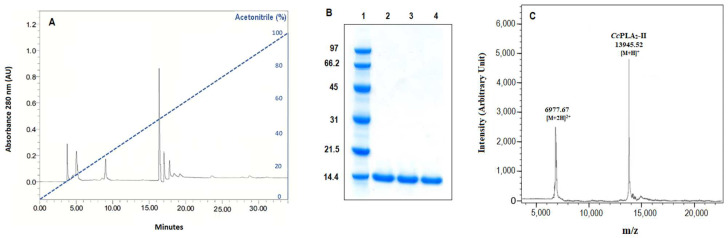
(**A**) Chromatography of RP-HPLC using a C18 column and (**B**) SDS-PAGE profile (15%) of purified Cc-PLA_2_-II. Lane 1: molecular mass markers, lanes 2 and 3: purified Cc-PLA_2_-II, land 4: Wa-PLA_2_-II. Blue line indicates linear acetonitrile gradient. (**C**) MALDI-TOF spectrum of Cc-PLA_2_-II.

**Figure 2 molecules-28-06517-f002:**
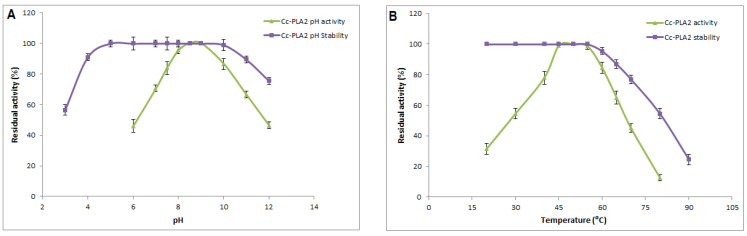
(**A**) pH profile and stability of purified Cc-PLA_2_-II. Phospholipase activity was measured at various pH levels ranging from 6.0 to 12.0 at optimal temperature (55 °C). The pH stability was determined by incubating purified Cc-PLA_2_-II in different buffers and at different pH values for 30 min at room temperature and residual enzyme activity was determined under standard assay conditions. (**B**) Effect of temperature on activity and stability of purified Cc-PLA_2_-II. Activity was measured and stability was quantified after enzyme was incubated at different temperatures (20 °C to 90 °C) by measuring the residual activity under standard conditions. Results represent means of three independent experiments and are expressed as mean ± SD (*n* = 3).

**Figure 3 molecules-28-06517-f003:**
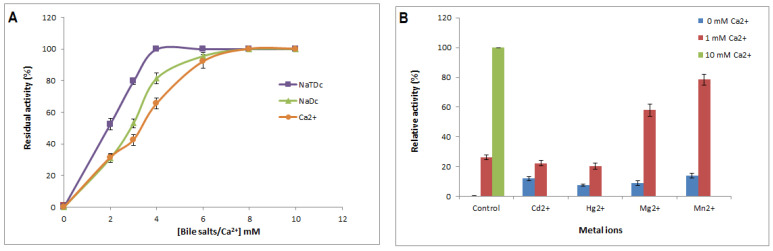
(**A**) Effect of calcium concentration on purified Cc-PLA_2_-II activity. Enzyme activity was measured at various concentrations of Ca^2+^ (0 to 10 mM) at pH 9 and 55 °C in the presence of 4 mM NaTDC. The absence of phospholipase activity in the presence of 10 mM EGTA/EDTA and the absence of Ca^2+^ is indicated by stars. (**B**) Effect of several metal ions on activity of Cc-PLA_2_-II. Influence of 10 mM metal ions on activity in the absence or presence of 1 mM Ca^2+^ was investigated in the presence of 4 mM NaTDC using PC emulsion as a substrate at 55 °C and pH 9. Control represents 100% of PLA_2_ activity at 8 mM Ca^2+^ under the same conditions. Results represent means of three independent experiments and are expressed as mean ± SD (*n* = 3).

**Figure 4 molecules-28-06517-f004:**
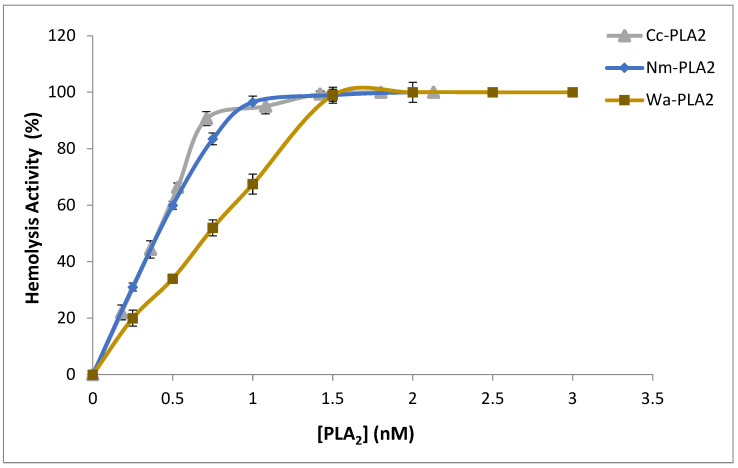
Indirect hemolytic activity of Cc-PLA_2_-II, Nm-PLA_2_, and Wa-PLA_2_ against human erythrocytes. Purified PLA_2_ in different concentrations (0 to 2.5 nM) were incubated at 37 °C for 10 min with PC, erythrocytes, and PBS (1:1:8 *v*/*v*) to release Hb. Absorbance at 540 nm allowed the determination of free Hb released. Results represent means of three independent experiments and are expressed as mean ± SD (*n* = 3). Hb; hemoglobin.

**Figure 5 molecules-28-06517-f005:**
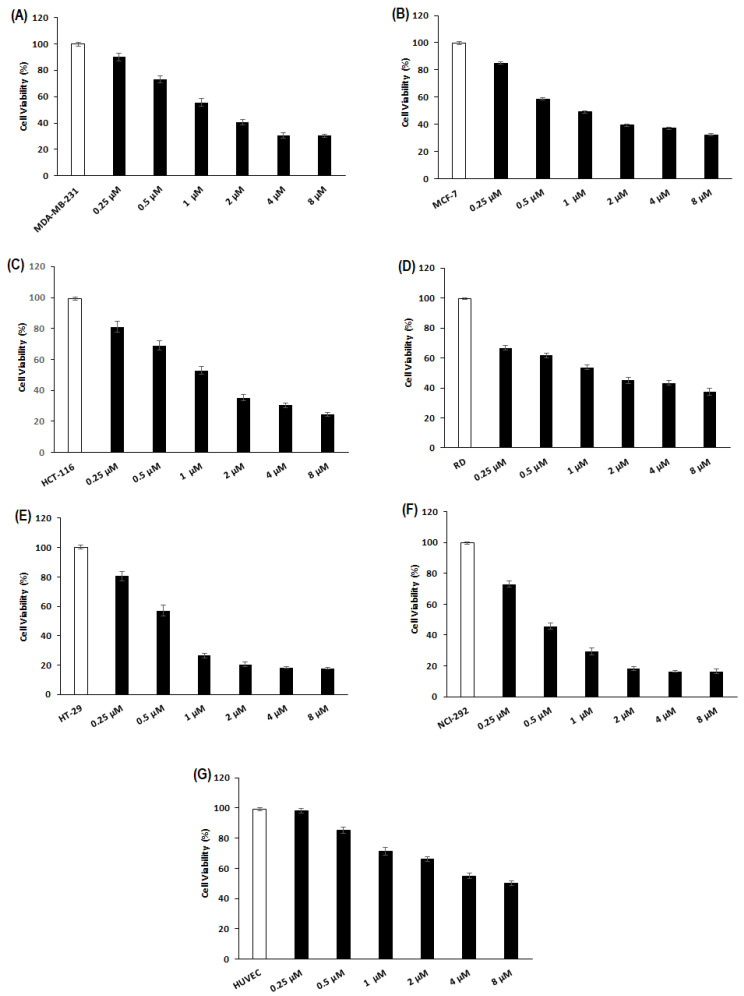
Cytotoxic effect of Cc-PLA_2_-II in normal and cancer cell lines. (**A**) MDA-MB231: human breast adenocarcinoma; (**B**) MCF-7: human breast-cancer; (**C**) HCT-116: human colorectal carcinoma; (**D**) RD: human rhabdomyosarcoma; (**E**) HT-29: adenocarcinoma; (**F**) NCI-292: mucoepidermoid carcinoma; (**G**) HUVECs: human umbilical-vein endothelial cells. All cells were treated with Cc-PLA_2_-II concentrations ranging from 0.25 to 8 µM. Cytotoxicity assay was conducted using MTT reagent. Results represent means of three independent experiments. Viability was calculated as mean ± SD (*n* = 3).

**Table 2 molecules-28-06517-t002:** *N*-terminal sequence alignment of Cc-PLA_2_-II from snake-venom sPLA_2_.

Species	PLA_2_	Sequence	% of Identity	Reference
*C. cerates*	Cc-PLA_2_-II	QFGKMIKLTTGKSAALSYSAYGCYCGWGGQGKPQDATDHCCFLH	-	
*C. cerates*	Cc-PLA_2_-I	QFGKMIKHKTGKSALLSYSAYGCYCGWGGQGKPQDATDHCCFVH	91%	[26]
*V. renardi*	Vur-PLA_2_	QFGKMIKYKTGKSALLSYSAYGCYCGWGGQGKPQDPTDRCCFVH	86%	[27]
*V. ursinii*	AmI2	QFGKMIKYKTGKIALFSYSDYGCYCGWGGQGKPKDATDRCCFVH	80%	[28]
*C. cerates*	Cc-PLA_2_-2	QFGKMIKHKTGKSALLSYSGNPCYCGWGGQGPPQDATDHCCFVH	82%	[26]

**Table 3 molecules-28-06517-t003:** Specific activities of Cc-PLA_2_-II, Wa-PLA_2_ and Nm-PLA_2_ on 4-nitro-3-(octanoyloxy) benzoic acid (4N3OBA).

	Cc-PLA_2_-II	Wa-PLA_2_-II	Nm-PLA_2_
Specific activity (IU/mg)	677.2 ± 11.2	481.6 ± 15.7	333.6 ± 10.6

## Data Availability

The data presented in this study are available on request from the corresponding author.

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
