# Peer review of "Functional Characterization and Anti-Tumor Effect of a Novel Group II Secreted Phospholipase A2 from Snake Venom of Saudi Cerastes cerates gasperetti"

_molecules, 2023, doi:10.3390/molecules28186517_

Round 1
Reviewer 1 Report
The article entitled “Functional characterization and anti-tumoral effect of a novel group II secreted Phospholipase A2 from snake venom of Saudi Cerastes discusses the characterization in terms of enzymatic characteristics, protein analysis, haemolytic activity and cytotoxicity assay. There are some concerns needed to be addressed.
1. The write-up's serious lack of experiment problem statement and hypothesis, especially the abstract and introduction. Although you have done the analysis, there is no strength in writing the abstract, which could appeal to the readers meaningfully.
2. revise the introduction and add why you did various analyses like enzyme characteristics, what is needed and the problem statement. Then, you can give info by using previous papers.
3. There is a serious hole in the anti-tumoral effect. You must do at least a few analyses to strengthen the postulate that they cause cytotoxicity. What is the underline mechanism or mode of cell death at least? Same as you mentioned in the discussion, the previous studies regarding angiogenesis or metastatic behaviour, but there is no assay conducted by you. This is a weak impression towards what you say about novel Cc-PLA2.
4. Can you clarify any novelty based on extracted product protein analysis, enzyme, or stability? You have to highlight some aspects that reflect your novel product's potential.
5. Invitro cell line study was only conducted for 24 hours which is not significant in terms of time. The study should be done at 72 hrs (3 days). In your results, you mention there is time and dose dependant cytotoxicity, which is incorrect as no time-wise experiment was done, only 24hr.
6. The lack of result discussion of cytotoxicity studies. As you have screened various cell lines, mention the sensitivity profile of which cell line is more sensitive towards CcPLA2 and so on… and compare a bit as All are different cell lines, so the mechanism of cytotoxicity should be different.
7. Similarity index 40 % (recheck, please) and rewrite
8. Do mention the ethical approval statement for the cell line study
improve the structure of intro and abstract
Author Response
- Reviewer 1:
The article entitled “Functional characterization and anti-tumoral effect of a novel group II secreted Phospholipase A2 from snake venom of Saudi Cerastes discusses the characterization in terms of enzymatic characteristics, protein analysis, hemolytic activity and cytotoxicity assay.
There are some concerns needed to be addressed.
Comment 1: The write-up's serious lack of experiment problem statement and hypothesis, especially the abstract and introduction. Although you have done the analysis, there is no strength in writing the abstract, which could appeal to the readers meaningfully.
- The abstract has been modified according to the reviewer’s comment (Please See Page 2).
Comment 2: Revise the introduction and add why you did various analyses like enzyme characteristics, what is needed and the problem statement. Then, you can give info by using previous papers.
- The introduction has been modified and a problem statement has been added as suggested (Please see the introduction section, page 4).
Comment 3: There is a serious hole in the anti-tumoral effect. You must do at least a few analyses to strengthen the postulate that they cause cytotoxicity. What is the underline mechanism or mode of cell death at least? Same as you mentioned in the discussion, the previous studies regarding angiogenesis or metastatic behaviour, but there is no assay conducted by you. This is a weak impression towards what you say about novel Cc-PLA2.
- Thank you very much for this interesting comment. Actually, in this study, we tested the toxicity of Cc-PLA2-II against several human cancer cell lines. Then, we compared this effect with previously studied group II snake venom PLA2 We also discussed the possible mechanism(s) of this enzyme without any experimental assay. Indeed, PLA2 cytotoxicity could due to diverse mechanisms that could be studied by:
* Flow cytometry to determine apoptosis cell pathways
* Generation of reactive oxygen species (ROS) during membrane phospholipid degradation determined by measuring the release of lactate dehydrogenase (LDH) activity
* DNA damage of treated cell determined by comet assay
These mechanisms will be experimentally investigated, analyzed and discussed in the future work.
Comment 4: Can you clarify any novelty based on extracted product protein analysis, enzyme, or stability? You have to highlight some aspects that reflect your novel product's potential.
Response to Reviewer comment N°4:
- The first point of novelty is the geographic localization of Cerastes cerastes Cc-PLA2-II is the first sPLA2 purified and well characterized from Saudi desert Cerastes cerastes. Particularly, its biochemical properties, especially pH and temperature stability, were attractive and different when compared to other PLA2 purified from Tunisian and Algerian Cerastes cerastes. In addition, the cytotoxicity profile of the purified enzyme against human cancer cells was totally different from those of CC-PLA2-1 and CC-PLA2-2 isolated from Tunisian Cerates cerastes which are not cytotoxic towards normal and cancer cells.
Comment 5: In vitro cell line study was only conducted for 24 hours which is not significant in terms of time. The study should be done at 72 hrs (3 days). In your results, you mention there is time and dose dependant cytotoxicity, which is incorrect as no time-wise experiment was done, only 24hr.
- MTT assay is used for determine cytotoxicity effect by incubating cells with corresponding molecule during 30 minutes to 24h. Incubation during 72h is generally realized when the molecule does not exhibit any effect on treated cells during 24h and then its action on proliferation is determined. These experiment details are used in the majority of works such as the following references:
1- Van Meerloo, J.; Kaspers, G. J.; Cloos, J., Cell sensitivity assays: the MTT assay. Cancer cell culture: methods and protocols 2011, 237-245.
2- Buch, K.; Peters, T.; Nawroth, T.; Sänger, M.; Schmidberger, H.; Langguth, P., Determination of cell survival after irradiation via clonogenic assay versus multiple MTT Assay-A comparative study. Radiation oncology 2012, 7, (1), 1-6.
Comment 6: The lack of result discussion of cytotoxicity studies. As you have screened various cell lines, mention the sensitivity profile of which cell line is more sensitive towards CcPLA2 and so on… and compare a bit as All are different cell lines, so the mechanism of cytotoxicity should be different.
- Data of cytotoxicity studies have been discussed and compared to previously reported snake venom sPLA2s against the same human cell lines. We have also discussed the potential mechanism of cytotoxicity in the last paragraph 2.3.2 page 14. After describing results about cytotoxicity, we have mentioned that NCI-292 and HT-29 are more sensible than MDA-MB- 231, MCF-7, HCT-116 and RD cell lines. We also added a possible explanation for this sensibility.
Please see the following sentence in paragraph 2.3.2 page 14 (Lines 19-24): “IC50 values of 3 µM were recorded for MDA-MB- 231, MCF-7, HCT-116 and RD (Figure 5.A-D). However, NCI-292 and HT-29 were the most sensible with IC50 values of 1.8 and 2.2 µM, respectively (Figure 5.E-F). The fact that cells’ damage is generally mediated by a specific interaction of PLA2 with cell surface receptors which are different between various cancer cells could explain the difference of sensibility between used cancer cell lines [49]. ”.
Since used cells lines have different profiles of sensitivity towards Cc-PLA2-II, each one could present different mechanism (s) of cytotoxicity. This point has also been indicated in the paragraph 2.3.2 page 15.
Comment 7: Similarity index 40 % (recheck, please) and rewrite
- The similarity index has been checked before submission (=17.4%) and after revision (= 7.7%) using PlagScan software (Please See the Attached files).
Comment 8: Do mention the ethical approval statement for the cell line study.
- The ethical approval statement for the cell line study has been added as recommended (Please see page 20).

Reviewer 2 Report
1. In Figure 1 (A), the identification of CcPLA2-II is missing, Figure (B) can be separated without the need to merge with (c).
2. In Figure 2, the identification of purple and green line segments is missing. In Figure 3, the abscissa is not appropriate.
3. Is Cc-PLA2-II a new protein? Please state it clearly.
4. In this article, mostly is about the experimental results and less is about the comparison with other proteins. Please add more about the advantages of Cc-PLA2-II and relevant analysis.
There are too many grammatical mistakes. The English is obscure. Please check all the paper carefully.
For example, p2,“a several pharmacological activities” should be “several pharmacological activities”.
MALDI-Tof should be all capitalized.
p4,“The ability of Cc-PLA2-II to hydrolyze the chromogenic substrate can suggests” that “should be “The ability of Cc-PLA2-II to hydrolyze the chromogenic substrate can suggest”.
“is normally able to attacks substrates in a micellar state…,” should be “is normally able to attack substrates in a micellar state…”.
p5, “ranging from 3 and 9” should be “ranging from 3 to 9”
Author Response
- Reviewer 2 :
Comment 1: In Figure 1 (A), the identification of CcPLA2-II is missing, Figure (B) can be separated without the need to merge with (c).
- As suggested by the reviewer, the identification of Cc-PLA2-II has been added in figure 1. Figure 1B has been separated from figure 1C (Please see figure 1, page 6).
Comment 2: In Figure 2, the identification of purple and green line segments is missing. In Figure 3, the abscissa is not appropriate.
- Figure 2 has been modified according the comment of the reviewer (Please see figure 2, page 10).
- In figure 3, the abscissa ([Bile salts/Ca2+]) indicates the corresponding concentrations of bile salts or calcium ranging from 0 to 12 mM.
Comment 3: Is Cc-PLA2-II a new protein? Please state it clearly.
- Cc-PLA2-II is a new member of group II secreted phospholipase A2. This statement has been mentioned in the abstract and the introduction as well as in the conclusion sections.
Please see:
- The abstract page 2: “Here, a new group II secreted phospholipase A2 (Cc-PLA2-II) from the snake venom of Saudi Cerastes cerastes gasperetti has been isolated and characterized. Purified enzyme showed a molecular weight of 13.945 kDa and a highest specific activity on emulsified phosphatidylcholine of 1560 U/mg at pH 9.5 and 50°C with a strict calcium dependence”.
- The last paragraph in the introduction section pages 4 and 5: “This study reported a functional characterization of a new snake venom secreted PLA2 purified from horned viper Cerastes cerastes gasperetti (Cc-PLA2-II) coming from Saudi Arabia desert. Interesting anti-tumoral effects on six human tumoral cell lines were described and compared to those of sPLA2 from Tunisian desert Cerastes cerastes venom. This biological activity of snake venom sPLA2 suggested that these enzymes have a crucial role in alteration of tumoral progression steps and can be potential targets for anti-cancer new drugs.”.
- The conclusion section page 26: “A new group II snake venom secreted PLA2 designed by Cc-PLA2-II was purified and characterized from desert cobra venom of Cerastes cerastes gasperettii coming from Saudi Arabia. It presented heterogenic and a selective antitumor profile against several human cancer-lines in a dose dependent manner. Further studies will be carried out to determine the cytotoxic mechanism of Cc-PLA2-II. ”
Comment 4: In this article, mostly is about the experimental results and less is about the comparison with other proteins. Please add more about the advantages of Cc-PLA2-II and relevant analysis.
- In each paragraph in the result and discussion section, we have compared each finding with previous findings on group II secreted PLA2 purified from snake venom. We have clearly described and discussed the fact that Cc-PLA2-II exhibited similar biochemical characteristics and biological effects towards human erythrocytes and tumoral cells.
Comment 5: Comments on the Quality of English Language
1-There are too many grammatical mistakes. The English is obscure. Please check all the paper carefully.
- The whole manuscript has been carefully revised. Some mistakes have been corrected.
2-For example, p2,“a several pharmacological activities” should be “several pharmacological activities”.
- “a” has been removed (Please see line 26, page 3).
3- MALDI-Tof should be all capitalized.
- The expression “MALDI-Tof” has been capitalized as MALDI-TOF (Please see line 10, page 6 and line 3, page 8).
4-p4,“The ability of Cc-PLA2-II to hydrolyze the chromogenic substrate can suggests” that “should be “The ability of Cc-PLA2-II to hydrolyze the chromogenic substrate can suggest”.
- It has been corrected (Please see line 8, page 8).
5-“is normally able to attacks substrates in a micellar state…,” should be “is normally able to attack substrates in a micellar state…”.
- It has been corrected (Please see line 14-16, page 8).
6-p5, “ranging from 3 and 9” should be “ranging from 3 to 9”
- It has been corrected (Please see line 13-14, page 9).

Reviewer 3 Report
In this manuscript, Alonazi et al. functionally characterized a group II secreted phospholipase A2 from snake venom of Saudi Cerastes cerates gasperetti, and its anti-tumor effects were also investigated. It can be accepted after revision.
1) Line 93, the paper describing the purification protocol, should be cited.
2) Table 3, the standard deviation was missing. Additionally, the statistical method should be described.
3) Figure 2, the authors need to describe what the blue and green lines stand for.
4) Figure 4, a control (eg. Buffer alone) is needed.
5) The legend of Figure 5, what does normal cell line mean?
English can be improved.
Author Response
Reviewer 3:
In this manuscript, Alonazi et al. functionally characterized a group II secreted phospholipase A2 from snake venom of Saudi Cerastes cerates gasperetti, and its anti-tumor effects were also investigated. It can be accepted after revision.
Comment 1: Line 93, the paper describing the purification protocol, should be cited.
- The purification protocol of Cc-PLA2-II has been detailed in materiel and methods section (Please see paragraph 3.2, page 17).
Comment 2: Table 3, the standard deviation was missing. Additionally, the statistical method should be described.
- Standard deviations have been added in table 3 in the revised manuscript (Please see table 3 page 10).
- A new paragraph describing the statistical analysis has been added in the revised manuscript (Please see page 19).
Comment 3: Figure 2, the authors need to describe what the blue and green lines stand for.
Response to Reviewer comment N°3:
Figure 2 has been modified in order to clarify what the blue and green lines stand for (Please see the figure 2 page 13).
Comment 4: Figure 4, a control (eg. Buffer alone) is needed.
- In the direct hemolytic assay, no hemoglobin was released when erythrocytes were incubated with buffer PBS alone. The percentage of hemolysis was, therefore, zero. For this reason, we did not present this result in figure 4.
Comment 5: The legend of Figure 5, what does normal cell line mean?
- In the caption of figure 5, HUVEC: human umbilical vein endothelial cells were used as non tumoral cells (normal cell lines).

Reviewer 4 Report
The manuscript titled (Functional characterization and anti-tumoral effect of a novel group II secreted Phospholipase A2 from snake venom of Saudi Cerastes cerates gasperetti) reported a functional characterization of a novel snake venom secreted PLA2 purified from horned viper Cerastes cerastes gasperetti coming from Saudi Arabia desert.
The manuscript is generally well-written and structured. The analysis was successful, and the data was well understood and modeled in detail. In addition, the manuscript contains relevant paragraphs that have been discussed. The selection of the bibliography is appropriate to the content of the manuscript.
However, some errors appeared throughout the manuscript.
1- There are no details on the different materials (origin of products, purchasers, etc.) used throughout the study (material and methods section)
2- The authors should verify if the sPLA2 used for comparison throughout the main text belongs to secreted PLA2 group II.
3- The authors should mention if the commercial Nm-PLA2 is a purified or a synthetic enzyme.
4- Please provide the concentration (on mM) of pc used in Cc-PLA2-II dosage.
5- The authors should explain briefly the relation between biochemical characteristics and anti-tumoral effect.
6- Results of direct hemolytic activity are not presented in Figure 4—also, no data about Nm-PLA2 and Wa-PLA2.
7- The authors should clarify the results described in section 2.3.1 on direct or indirect hemolytic effects.
8- The abbreviations used in the paper should be first spelled out and double-checked. Please carefully check.
9- Please remove or change the personal words "our" and "we" in the whole text with another expression.
10- There are some mistakes in the reference section. Please re-check and consult the updated guides to authors, which are available on this journal's website, to make them fully comply with the requested style and format of the journal.
Author Response
Reviewer 4 :
The manuscript titled (Functional characterization and anti-tumoral effect of a novel group II secreted Phospholipase A2 from snake venom of Saudi Cerastes cerates gasperetti) reported a functional characterization of a novel snake venom secreted PLA2 purified from horned viper Cerastes cerastes gasperetti coming from Saudi Arabia desert.
The manuscript is generally well-written and structured. The analysis was successful, and the data was well understood and modeled in detail. In addition, the manuscript contains relevant paragraphs that have been discussed. The selection of the bibliography is appropriate to the content of the manuscript.
However, some errors appeared throughout the manuscript.
Comment (1): There are no details on the different materials (origin of products, purchasers, etc.) used throughout the study (material and methods section)
- As suggested by reviewer, we have added, in the revised manuscript, a new paragraph (1 Reagents) showing details on the different materials (Please see paragraph 3.1, lines 9-22, page 17).
“Reagents
Chemicals were obtained from commercial sources. Chromatography material (Reverse phase High performance liquid chromatography RP-HPLC), phosphate buffer saline (PBS, pH 7.4), ethanol, acetonitrile, Trifluoroacetic acid TFA, Sodium dodecyl sulfate (SDS), acrylamide, ammonium persulfate, N,N,N’,N’-tetramethyl ethylenediamine (TEMED), β-mercaptoethanol and Coomassie brilliant blue R-250 were obtained from Bio-Rad.
Phosphatidylcholine (PC), 4-nitro-3-octanoyloxybenzoic acid (4N3OBA) , NaCl, CaCl2, Tris–HCl, Bovine serum albumin (BSA), sodium taurodeoxycholate (NaTDC), sodium deoxycholate (NaDC), ethylenediaminetetraacetic acid (EDTA), Bovine serum albumin (BSA), Triton X100, proteins markers for molecular masses, commercial PLA2 from Naja mossambica mossambica (Nm-PLA2, Sigma P7778) were purchased from Sigma Aldrich (St. Quentin-Fallavier, France).
Dulbecco's Modified Eagles Medium, calf serum fetal, glutamine, Streptomycin, Penicillin and 3-[4,5-dimethylthiazol-2-yl]-2,5 diphenyl tetrazolium bromide (MTT) were purchased from Life Technologies, Paisley, UK.”.
Comment (2): The authors should verify if the sPLA2 used for comparison throughout the main text belongs to secreted PLA2 group II.
- All secreted PLA2 used for comparison in results and discussion section belong to group II.
Comment (3): The authors should mention if the commercial Nm-PLA2 is a purified or a synthetic enzyme.
- Nm-PLA2 used as positive control in our study is a commercial PLA2 from Naja mossambica mossambica venom (Nm-PLA2, Sigma P7778). It is purified from Naja mossambica mossambica venom and commercialized as lyophilized powder.
https://www.sigmaaldrich.com/TN/en/product/sigma/p7778
Comment (4): Please provide the concentration (on mM) of pc used in Cc-PLA2-II dosage.
- The concentration of PC corresponds approximately to 13 mM using an average molecular mass of 770 g/mol for PC. This information has been added in the revised version (Please see page 5 (line 15) and page 8 (Line 16)).
Comment (5): The authors should explain briefly the relation between biochemical characteristics and anti-tumoral effect.
- Enzymes’ biochemical characteristics especially activity and stability at pH and temperatures as well as effect of bivalent cations provide information on the enzyme half time and conditions of catalytic action. Accordingly, we can decide the incubation time of the studied enzyme with tumoral cells and we will be sure that the culture cells medium composition (such as bivalents cations, growth factors, peptides…) and physic-chemical parameters like pH and temperature will not affect the enzymatic action toward tumoral cells. Therefore, biochemical characteristics determination help to better discuss and explain the mechanism of enzyme on the tumoral cells.
Comment (6): Results of direct hemolytic activity are not presented in Figure 4 also, no data about Nm-PLA2 and Wa-PLA2.
Response to Reviewer comment N°6:
- Figure 4 showed the indirect hemolytic effect of the studied Cc-PLA2-II. This result was presented and discussed in paragraph 2.3.1 (Hemolytic activity) (Please see page 13).
- Results of indirect hemolytic activity of Wa-PLA2 and Nm-PLA2 have been added in figure 4 and discussed in paragraph 2.3.1 (Hemolytic activity) (Please see page 13).
Comment (7): The authors should clarify the results described in section 2.3.1 on direct or indirect hemolytic effects.
- As suggested by the reviewer, we have clarified the results about direct and indirect hemolytic activity of Cc-PLA2-II, Nm-PLA2 and Wa-PLA2. Indeed, the three PLA2 exhibited indirect hemolytic activity. No direct hemolytic activity was detected for all tested enzymes. We apologize for the mistake about the direct hemolytic activity (Please see figure 4 and paragraph 2.3.1 (Hemolytic activity) page 13).
Comment (8): The abbreviations used in the paper should be first spelled out and double-checked. Please carefully check.
- All abbreviations used in the main text are now first spelt out and double-checked.
Comment (9): Please remove or change the personal words "our" and "we" in the whole text with another expression.
- As suggested by the reviewer, the personal words "our" and "we" have been removed.
Comment (10): There are some mistakes in the reference section. Please re-check and consult the updated guides to authors, which are available on this journal's website, to make them fully comply with the requested style and format of the journal.
- All references have been checked according the style and format of the journal.

Round 2
Reviewer 1 Report
Accepted
Author Response
Thank you for your input to improve and then accept our manuscript.
Reviewer 2 Report
The paper has been modified according to the previous review. In general, the Cc-PLA2-II in the article is lack of innovation, and there are not many comparisons between Cc-PLA2-II and old proteins, which does not reflect the advantages of Cc-PLA2-II.
The article is lack of the deep discussion and analysis of the results.
There are still many grammatical mistakes.
Like, in the abstract, "was observed through a cytotoxicity with " should be "was observed through cytotoxicity with "...
Author Response
Dear editor
We would like to thank the editor and the reviewer #2 for their thoughtful re-review of the manuscript. They raise important issues and their inputs are very helpful for improving the manuscript. We agree with their comments and we have revised our manuscript accordingly. The corresponding changes and refinements made in the revised paper are marked in red and summarized in our response below. English language editing of the revised manuscript will be arranged by MDPI services.
Reviewer 1:
The paper has been modified according to the previous review. In general, the Cc-PLA2-II in the article is lack of innovation, and there are not many comparisons between Cc-PLA2-II and old proteins, which does not reflect the advantages of Cc-PLA2-II.
- We agree perfectly with the reviewer comment. The cytotoxic effect section has been revised and many statements have been added to better discuss this section as recommended. Three new references have been also inserted in the revised version (Please see pages 14-16, 20 and 24).
There are still many grammatical mistakes.
- English language editing of the revised manuscript will be arranged by MDPI services.